# Challenging Low Homophily in Social Recommendation

## Abstract

Social relations are leveraged to tackle the sparsity issue of user-item interaction data in recommendation under the assumption of social homophily. However, social recommendation paradigms predominantly focus on homophily based on user preferences. While social information can enhance recommendations, its alignment with user preferences is not guaranteed, thereby posing the risk of introducing informational redundancy. We empirically discover that social graphs in real recommendation data exhibit low preference-aware homophily, which limits the effect of social recommendation models. To comprehensively extract preference-aware homophily information latent in the social graph, we propose **S**ocial **H**eterophily-**a**lleviating **Re**wiring (SHaRe), a data-centric framework for enhancing existing graph-based social recommendation models. We adopt Graph Rewiring technique to capture and add highly homophilic social relations, and cut low homophilic (or heterophilic) relations. To better refine the user representations from reliable social relations, we integrate a contrastive learning method into the training of SHaRe, aiming to calibrate the user representations for enhancing the result of Graph Rewiring. Experiments on real-world datasets show that the proposed framework not only exhibits enhanced performances across varying homophily ratios but also improves the performance of existing state-of-the-art (SOTA) social recommendation models.

***Keywords:*** Social Recommendation, Graph Rewiring, Contrastive Learning, Data-centric AI

## 1 INTRODUCTION

With the proliferation of online media content, recommender systems have become pivotal for content filtering and improving user experience. Given that user-item interactions can be formed as bipartite interaction graphs, graph neural networks (GNNs) are utilized to enhance the effectiveness of traditional recommender systems [10, 25, 26]. Nevertheless, such interaction data is often sparse in real-world scenarios and recommender systems contend with the challenge of information insufficiency [11, 20]. To alleviate this, social graphs (or networks) are incorporated into recommender systems. This premise is rooted in the concept of *social homophily* [17], wherein users tend to form connections with individuals who share similar interests. Consequently, these connections among like-minded users are harnessed to compensate for the information scarcity in the interaction graph. Therefore, recommender systems can better discern

the preferences of users who have limited interactions with items, thereby delivering more personalized recommendations [5, 6, 30, 31, 34, 35].

Although social graphs have shown promise in mitigating sparsity issues within interaction graphs and enhancing the performance of recommender systems, the relationship between social homophily and user preferences regarding items remains an under-investigated area, particularly in light of the intricate topological structures inherent in graph-based representations [21, 32, 33]. Since social information does not explicitly encapsulate user preferences and their interactions with items, it can serve only as supplementary data for recommendations. This implies that not all social information is inherently reliable. Furthermore, the inclusion of a social graph might introduce redundant information [16, 36]. To evaluate the user preferences inherent in the social graph, we calculate the preference-aware homophily ratios (see definitions in Sec. 2.1) across three real and widely used social recommendation datasets: LastFM [33], Douban [35] and Yelp [38], and observe that the user connections in the social graph are not highly homophilic. As shown in Fig. 1, all the distributions of edge-wise homophily ratios in these three datasets approximately follow the power-law distribution, and the edge-wise homophily ratios of most user-user edges are close to 0. In addition, we show the graph-wise homophily ratio $\mathcal{H}_s$ of each dataset, the result indicates that all the social graphs of these three datasets are low homophilic (ratios around $\mathcal{H}_s = 0.1$ or smaller). Consequently, these observations suggest that users connected in the social graph exhibit diverse preferences and the social graph is low homophilic (or heterophilic).

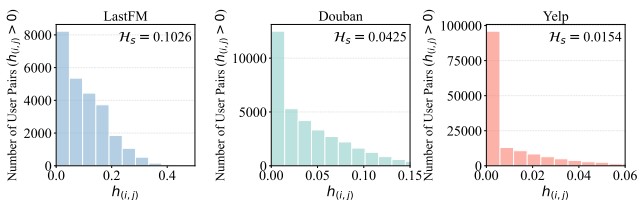

**Figure 1.** Preference-aware homophily ratio distributions of social graphs on three real-world datasets, where $h_{(i,j)}$ is the edge-wise homophily ratio of user-user edge $(u_i, u_j)$, $\mathcal{H}_s$ is the graph-wise homophily ratio of the social graph (see definitions in Section 2.1).

While limited preference-aware homophily is presented in the observation, current graph-based social recommendation models directly integrate the original social graph into the

system [14, 22, 30, 31], or construct the hypergraph based on the original social graph [1, 35]. Without a meticulous distinction between connections within the social graph, these approaches run the risk of permitting unreliable social relations to impact the recommendation process. Consequently, we contend that these social recommendation models do not fully harness the potential of the social graph and may not achieve optimality. To assess the effect of preference-aware homophily on social recommendation, we conduct experiments using two state-of-the-art models (SOTA), DiffNet [31] and MHCN [35] on synthetic sub-graphs with different graph-wise homophily ratios generated from LastFM dataset (see experiment settings in Sec. 4.1). As shown in Fig. 2, the performances of DiffNet and MHCN improve with increasing values of $\mathcal{H}_s$. This experimental result suggests that a higher graph-wise homophily ratio can consistently enhance the recommendation accuracy of social recommendation models. Given these findings, the key question emerges for social recommendations: *how can we optimize the utilization of reliable social relations and alleviate the unreliable relations to enhance recommendation performance?*

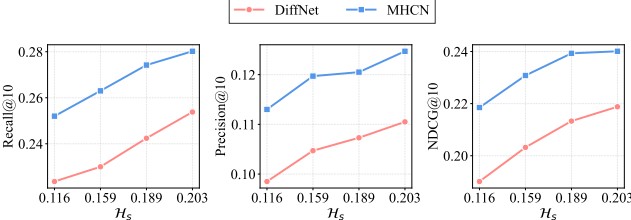

**Figure 2.** The influence of graph-wise homophily ratio to different social recommendation models.

In this paper, we aim to exhaustively explore the preference-aware homophily within social graphs and design a general framework for existing graph-based social recommendation models, enabling them to more adeptly discern user preferences and enhance their recommendation results. To this end, we propose a data-centric method to retain reliable social relations by rewiring the structure of the social graph, which can seamlessly integrate with any graph-based social recommendation model in a plug-and-play manner. Technically, we evaluate the user similarities based on user representations learned from the interaction graph. We then leverage these similarities to cut unreliable edges and add highly homophilic edges to the original social graph, thereby reconstructing a new graph for recommendation. However, altering the social graph structure through rewiring carries the risk of introducing noise, which can be exacerbated during model training. To mitigate this potential issue and improve both the confidence and effectiveness of the results yielded by Social Graph Rewiring, we integrate an innovative Homophilic Relation Augmentation (HRA) method achieved through the contrastive learning task. The positive and negative samples

are thoughtfully selected based on the edge-wise homophily ratios in the original social graph data and HRA refines the user representation by maximizing the consistency between the representation of the user and its similar users. As training progresses, user representations can be gradually calibrated through the integration of original social graph information, which then helps to enhance rewiring results.

The main contributions of this paper can be outlined as follows:

- **New Problem and Insights:** We delve into a critical yet relatively unexplored issue: low preference-aware homophily in social recommendation and to the best of our knowledge, we are the pioneers in exploring this challenge in social recommendation, where existing methods cannot explore the complicated social connections.
- **Innovative Methodology:** We propose a data-centric framework, which emphasizes retaining reliable social relations through Social Graph Rewiring and Homophilic Relation Augmentation. The social graph is rewired by cutting unreliable edges and adding highly homophilic edges. Besides, we integrate the Homophilic Relation Augmentation to further improve the outcomes of Social Graph Rewiring by enhancing the representations of users. Particularly, this framework is adaptable to any graph-based social recommendation model.
- **SOTA Performance:** We conduct extensive experiments on three public real-world datasets and SOTA social recommendation models. Experimental results verify that our framework can consistently improve the vanilla versions of SOTA methods, even under different graph-wise homophily ratios.

## 2 PRELIMINARIES

### 2.1 Definitions

In this section, we first define the important concepts used throughout the paper, then mathematically formulate the preference-aware homophily ratio, and finally define the research problem.

**Definition 1: Interaction Graph and Social Graph.** Let $\mathcal{U} = \{u_1, u_2, \ldots, u_m\}$ ($|\mathcal{U}| = m$) and $\mathcal{V} = \{v_1, v_2, \ldots, v_n\}$ ($|\mathcal{V}| = n$) denote the user set and item set, respectively, where $|\cdot|$ is the number of elements in the set. Given the user-item interaction matrix $R \in \mathbb{R}^{m \times n}$, we build a bipartite interaction graph $\mathcal{G}_r = (\mathcal{U} \cup \mathcal{V}, R)$, where $R_{u,v} = 1$ if user $u$ has interacted with item $v$ and $R_{u,v} = 0$ otherwise. Meanwhile, we denote the user social relations matrix as $S \in \mathbb{R}^{m \times m}$ and form the user social graph $\mathcal{G}_s = (\mathcal{U}, S)$.

**Definition 2: Preference-aware Homophily Ratio.** Given the subsets of items $\mathcal{V}_{u_i}$ and $\mathcal{V}_{u_j}$ that have been interacted with by users $u_i$ and $u_j$, respectively, we define the edge-wise homophily ratio based on Jaccard similarity [12]:

$$h_{(i,j)} = \frac{|\mathcal{V}_{u_i} \cap \mathcal{V}_{u_j}|}{|\mathcal{V}_{u_i} \cup \mathcal{V}_{u_j}|}. \tag{1}$$

$h_{(i,j)} \in [0, 1]$ stands for how similar the preferences of these two users are. Note that the user-user edge with strong homophily has a high homophily ratio $h_{(i,j)} \to 1$, which means they are more similar. In addition, we set $h_{(i,j)} = 0$ when only user $u_i$ or user $u_j$ appears in the training set. Based on $h_{(i,j)}$, we define the graph-wise homophily ratio:

$$\mathcal{H}_s = \frac{1}{N} \sum_{(i,j) \in \{S_{(i,j)}=1\}} h_{(i,j)}, \tag{2}$$

where $N$ is the number of edges in the social graph. $\mathcal{H}_s$ reflects the homophily of the holistic social graph.

**Research Problem Formulation:** Given the social graph $\mathcal{G}_s$, bipartite interaction graph $\mathcal{G}_r$, the main research problem of this paper is how to design a framework that can optimize the use of reliable social relations in the social graph and mitigate the impact of unreliable relations to improve recommendation performance.

## 2.2 Graph-based Social Recommendation Model

We utilize graph-based social recommendation models as the *backbone*, and we can easily plug our framework into it. Graph-based social recommendation models typically learn user latent factors from the social graph and user-item interaction graph, the information propagation layer for aggregating user and item representations can be defined as:

$$P^{(l+1)} = D_R^{-1} R Q^{(l)} + D_S^{-1} S P^{(l)}, \\ Q^{(l+1)} = D_{Rt}^{-1} R^{\mathrm{T}} P^{(l)}, \tag{3}$$

where $P^{(l)} \in \mathbb{R}^{m \times d}$ and $Q^{(l)} \in \mathbb{R}^{n \times d}$ are the user embedding and item embedding in layer $l$, respectively. $D_R$, $D_{Rt}$ and $D_S$ are the degree matrices of $R$, $R^{\mathrm{T}}$ and $S$. Based on the output user and item embeddings rows $p_u$ and $q_v$ from $P$ and $Q$, the predicted score $\hat{r}_{u,v}$ between user $u$ and item $v$ can be defined as $\hat{r}_{u,v} = p_u^\top q_v$. The Bayesian Personalized Ranking (BPR) loss is utilized to optimize the model [19]:

$$\mathcal{L}_{\mathrm{rec}} = \sum_{v \in \mathcal{V}_u, w \notin \mathcal{V}_u} -\log \sigma \left( \hat{r}_{u,v} - \hat{r}_{u,w} \right), \tag{4}$$

where $\sigma(\cdot)$ is the sigmoid function, $\hat{r}_{u,w}$ is the score of the sampled negative item $w$ without interaction with $u$.

## 3 METHODOLOGY

We design our **S**ocial **H**eterophily-**a**lleviating **Re**wiring (SHaRe) method, which is a data-centric framework that can be easily plugged in social recommendation models for handling the low homophily problem in social recommendation. The overall framework of SHaRe is shown in Fig. 3.

## 3.1 Social Graph Rewiring

In this part, we first introduce the computation of user similarity based on the learned user embeddings, and show the detail of the Social Graph Rewiring (SGR) process in SHaRe. In the training data, the presence of cold-start users and unseen users poses a challenge in SGR. We cannot calculate the edge-wise homophily ratio between some users as the similarity for graph rewiring, thus we adopt an encoder to learn the user embeddings for computing user similarities. Given the initial item embedding $q_v^{(0)}$, the user domain convolutional layer of the encoder can be defined as:

$$z_u^{(l+1)} = \sum_{v \in \mathcal{N}_u} \frac{1}{\sqrt{|\mathcal{N}_u|} \sqrt{|\mathcal{N}_v|}} q_v^{(l)}, \tag{5}$$

where $z_u$ is the output user embedding, we can concatenate all the $z_u$ as $Z$. Note that $z_u$ is used for the computation of the similarity, but it is not involved in the backbone model for recommendation. This user embedding only contains information learned from the interaction graph, and does not incorporate information from the social graph. We aim to measure the similarity of preferences between users, according to the information of users in the interaction graph. Therefore, we use the user embeddings to compute the cosine similarity between users $u_i$ and $u_j$:

$$\begin{aligned} c_{(i,j)} &= \phi(z_{u_i}, z_{u_j}) \\ &= \frac{z_{u_i} \cdot z_{u_j}}{\|z_{u_i}\| \cdot \|z_{u_j}\|}, \end{aligned} \tag{6}$$

where $\| \cdot \|$ denotes the vector norms.

Based on the similarity $c_{(i,j)}$ for each user-user pair, we rewire the original social graph by adding highly homophilic edges and cutting redundant edges.

As shown in Fig. 4, given the original edges set $\mathcal{S}$, we cut the existing edges which contain negative $c_{(i,j)}$ and $c_{(i,j)} = 0$:

$$\mathcal{S}_{\mathrm{cut}} = \{c_{(i,j)} | c_{(i,j)} \in \mathcal{S}, \ c_{(i,j)} <= 0\}. \tag{7}$$

The remaining edges set is:

$$\mathcal{S}_{\mathrm{remain}} = \mathcal{S} - \mathcal{S}_{\mathrm{cut}}. \tag{8}$$

New edges are added based on the Top-$M$ similarities:

$$\mathcal{S}_{\mathrm{add}} = \{c_{(i,j)} | \text{ Top-}M(c_{(i,j)})\}, \tag{9}$$

where $M$ is the number of cut edges based on Eq. (7). Note that the number of edges we add empirically in the social graph is the same as the number we cut, thereby ensuring the highest efficiency of SGR while maintaining the performance of SHaRe. Top-$M(c_{(i,j)})$ is the function that selects top $M$ similarities $c_{(i,j)}$.

Based on $\mathcal{S}_{\mathrm{remain}}$ and $\mathcal{S}_{\mathrm{add}}$, the rewired edges set is $\widehat{\mathcal{S}} = \mathcal{S}_{\mathrm{remain}} \cup \mathcal{S}_{\mathrm{add}}$, and the adjacency matrix of rewired graph $\widehat{\mathcal{G}}_s$ can be:

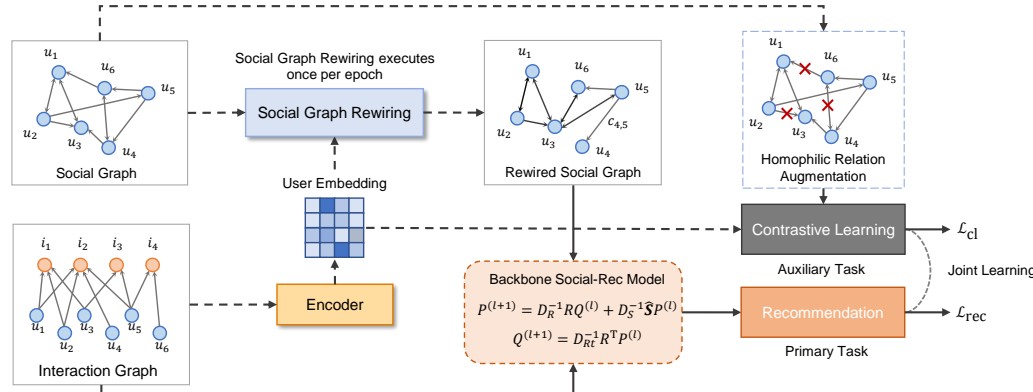

**Figure 3.** An overview of the proposed SHaRe framework. A recommendation encoder learns user embeddings $P$ from the interaction graph. These user embeddings $P$ are used for rewiring the social relations matrix $S$. The rewired social relations matrix $\widehat{S}$ and the interaction matrix $R$ are inputted to the backbone social recommendation models. Their output user and item representations are used for calculating the recommendation loss $\mathcal{L}_{\text{rec}}$.

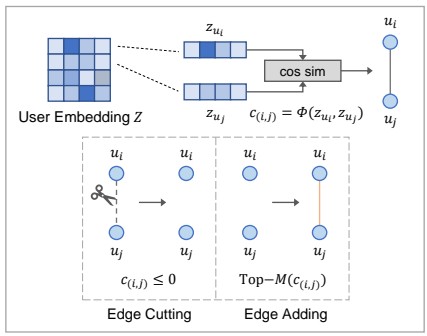

**Figure 4.** User embedding similarity and Social Graph Rewiring (SGR).

$$\widehat{S}_{(i,j)} = \begin{cases} \tilde{c}_{(i,j)}, & \text{if } c_{(i,j)} \in \widehat{\mathcal{S}} \\ 0, & \text{otherwise,} \end{cases} \quad (10)$$

where $\widehat{S}_{(i,j)}$ represents the social weight of the edge in the rewired graph, $\tilde{c}_{(i,j)}$ is the normalized cosine similarity.

### 3.2 Homophilic Relation Augmentation-based Joint Learning

Although SGR can extract the preference-aware homophily information in the social graph, simply rewiring the social graph brings the risk of introducing noise. To alleviate this issue and enhance the quality of the SGR result, we innovatively integrate the Homophilic Relation Augmentation (HRA) method to optimize the user embeddings $Z$. HRA uses information from the original social graph to select positive and negative samples based on edge-wise homophily ratio, and maximizes the consistency between positive sample users to further refine and calibrate the user embeddings $Z$, improving the SGR result during the training process.

We first construct the self-supervised signals by sampling highly homophilic user pairs to establish augmented views of users. We select the user pairs with high edge-wise

homophily ratios as the positive pairs and let the low homophilic pairs be negative pairs. As described in Sec. 3.1, given the output user embedding $Z$ learned by the encoder, we select the positive samples that are greater than a threshold $\epsilon$:

$$\mathcal{P}_{u_i+} = \{z_{u_k} \mid k \in \{h_{(i,k)} > \epsilon\}\}$$
$$\text{with } \epsilon = (\zeta - h_{\min}) \cdot (h_{\max} - h_{\min}), \quad (11)$$

where $h_{\max}$ and $h_{\min}$ are the maximal and minimum values among the homophily ratios. As observed in the edge-wise homophily ratio distributions in Fig. 1, the distributions are different in different datasets, which leads to different thresholds to filter high and low homophily ratios for different datasets. Therefore, we normalize the edge-wise homophily ratios $h_{(i,j)}$ and set a parameter $\zeta$ whose range is in $[0, 1]$ to better control the threshold $\epsilon$ for sampling.

With the positive samples $\mathcal{P}_{u_i+}$, we follow the previous research to utilize InfoNCE [18] as the contrastive learning loss, which is effective in estimating mutual information:

$$\mathcal{L}_{\text{cl}} = \sum_{u_i \in \mathcal{U}} -\log \frac{\sum_{p \in \mathcal{P}_{u_i+}} \Phi(z_{u_i}, z_{u_p})}{\sum_{p \in \mathcal{P}_{u_i+}} \Phi(z_{u_i}, z_{u_p}) + \sum_{u_j \in U/\mathcal{P}_{u_i+}} \Phi(z_{u_i}, z_{u_j})}, \quad (12)$$

where $\Phi(z_{u_i}, z_{u_p}) = \exp\left(\phi(z_{u_i} \cdot z_{u_p})/\tau\right)$, $\phi(\cdot)$ is the cosine similarity of two embeddings, $\tau$ is the temperature. We empirically set $\tau = 0.1$; for any given user $u_i$, we aim to encourage the coherence between the user embedding $z_{u_i}$ and the highly homophilic user embedding $z_{u_p}$ derived from $\mathcal{P}_{u_i+}$, and diminish the correlation between the user embedding $z_{u_i}$ and low homophilic user embedding $z_{u_j}$. This objective can be accomplished by contrastive learning.

In SHaRe, we adopt a joint learning strategy to optimize the model, including two tasks: Top-N item recommendation and HRA, while the recommendation is the primary task and

the contrastive learning is an auxiliary task. The joint loss of SHaRe is defined as:

$$\mathcal{L} = \mathcal{L}_{\text{rec}} + \lambda \mathcal{L}_{\text{cl}}, \tag{13}$$

where $\lambda$ is a parameter used for constraining the magnitude of contrastive learning. The sensitivity of $\lambda$ is analyzed in Sec. 4.6.

### 3.3 Overall Process of SHaRe Framework

The overall process of the SHaRe framework is shown in Algorithm 1. Line 3 is the computation of a recommendation encoder; Lines 4 to 22 are for the processes of SGR and the computation of the backbone social recommendation models (see detail in Sec. 2.2); Lines 23 to 26 are for the process of objective losses computations and joint learning. Particularly, to achieve the best performance and improve efficiency for our framework, we adopt a warm-up strategy and only execute SGR once at the start of each epoch (Lines 4 to 5). As shown in the Algorithm 1, we start executing SGR from the 10th epoch as a warm-up, and only execute it during the first training iteration of the epoch. The analysis of effectiveness and efficiency indicates that these strategies make SHaRe more efficient and effective (see details in Sec. 4.4). Once we execute SGR, the new rewired social relations matrix $\widehat{S}_{\text{new}}$ is updated and is inputted to train the backbone social recommendation models for computing user and item embeddings (Lines 17 to 18). If SGR has not started execution yet, we then input the original social relations matrix $S$ to the backbone (Line 22).

## 4 EXPERIMENTS

In this section, we design the experiments to evaluate our SHaRe framework and answer the following research questions: **RQ1** How do the social recommendation models enhanced by SHaRe perform compared to the vanilla version? **RQ2** How does the SHaRe perform under different graph-wise homophily ratios? **RQ3** What is the benefit of different SGR strategies in SHaRe? **RQ4** How do the components and the different operations of SGR affect the performance of SHaRe? **RQ5** How do the parameters affect SHaRe?

### 4.1 Experimental Settings

**4.1.1 Datasets.** We utilize three real-world datasets: LastFM[1], Douban[2] and Yelp[3] in our experiments to evaluate the effectiveness of SHaRe. Table 1 shows the detailed statistics of the datasets. The user ratings of the Douban and Yelp datasets scale from 1 to 5. Following [35] and [33], we remove interactions with ratings less than 4, aiming to better improve Top-N recommendation. We split the datasets into a training set, validation set, and test set with a ratio of 8:1:1.

---

[1]http://files.grouplens.org/datasets/hetrec2011/

[2]https://pan.baidu.com/s/1hrJP6rq

[3]https://github.com/Sherry-XLL/Social-Datasets/tree/main/yelp2

---

**Algorithm 1:** The overall process of SHaRe.

**Input:** Social graph $\mathcal{G}_s(\mathcal{U}, S)$; interaction graph $\mathcal{G}_r = (\mathcal{U} \cup \mathcal{V}, R)$; sampling parameter $\zeta$; contrastive learning coefficient $\lambda$; backbone social recommendation model: backbone($\cdot$) (see definition in Sec. 2.2).

**Output:** Optimal user and item embeddings; optimal model parameters.

1   **for** $k = 1, \ldots, K$ **do**
2    **for** $n = 1, \ldots, N$ **do**
3     $Z = \text{encoder}(R)$;
4     **if** $k >= 10$ **then**
5      **if** $n = 1$ **then**
6       $M = 0$;
7       **for** $z_{u_i} \in Z$ **do**
8        **for** $z_{u_j} \in Z(j \neq i)$ **do**
9         $c_{(i,j)} = \phi(z_{u_j}, z_{u_j})$;
10         **if** $c_{(i,j)} <= 0$ **then**
11          Cut the edge $(u_i, u_j)$ from $\mathcal{G}_s$ with Eq. (7);
12          Remain edges with Eq. (8);
13         $M + 1$;
14       Add new edges with Eq. (9);
15       $\tilde{c}_{(i,j)} = (c_{(i,j)} - c_{\min}) \cdot (c_{\max} - c_{\min})$;
16       Build $\widehat{S}$ with Eq. (10);
17       $\widehat{S}_{\text{new}} := \widehat{S}$;
18       $P, Q = \text{backbone}(\widehat{S}_{\text{new}}, R)$;
19      **else**
20       $P, Q = \text{backbone}(\widehat{S}_{\text{new}}, R)$;
21     **else if** $k < 10$ **then**
22      $P, Q = \text{backbone}(S, R)$;
23     Calculate $\mathcal{L}_{\text{rec}} = \text{BPR}(P, Q)$;
24     Select positive samples $\mathcal{P}_{u_i+}$ with Eq. (11) in terms of $\zeta$;
25     Calculate $\mathcal{L}_{\text{cl}} = \text{InfoNCE}(\mathcal{P}_{u_i+}, Z)$;
26     Jointly optimize overall objectives with Eq. (13);
27   **return** Optimal user and item embeddings.

---

**Table 1.** Statistics of datasets.

| Dataset | User | Item | Feedback | Relation | Density | $\mathcal{H}_s$ |
|---|---|---|---|---|---|---|
| LastFM | 1,892 | 17,632 | 92,834 | 25,434 | 0.28% | 0.1026 |
| Douban | 2,848 | 39,586 | 894,887 | 35,770 | 0.79% | 0.0425 |
| Yelp | 16,239 | 14,284 | 198,397 | 158,590 | 0.08% | 0.0154 |

To conduct the experiment that validates the influence of homophily ratios, we first generate synthetic social subgraphs with different graph-wise homophily ratios $\mathcal{H}_s$ using the method similar to [40]. Moreover, we control the number

of users in the range [590, 600] for the consistency of experimental settings. Corresponding to the social sub-graph, we set the same number of users in the sub-interaction graph.

**4.1.2 Baselines.** We compare four vanilla state-of-the-art social recommendation models with the version enhanced by our framework:

- **DiffNet [31]:** It utilizes an effective fusion layer to refine the user embeddings by incorporating a social influence diffusion process.
- **DiffNet++ [30]:** It is the improved model of DiffNet that aggregates the social diffusion in both the user and item domains.
- **LightGCN+social:** It is implemented by adding social diffusion components into the user embedding aggregation layer of LightGCN [10] for social recommendation.
- **MHCN [35]:** It uses multi-channel triangular motifs to construct social hypergraph for modeling social recommender system, which models the complex social relations of users to improve effectiveness.

To validate the effectiveness of SHaRe, the enhanced baselines are compared with all their vanilla versions. Besides, since MHCN is the latest and most efficient SOTA method in social recommendation, it is used as the backbone of the SHaRe framework in the ablation study and parameter sensitivities analysis of our experiment.

**4.1.3 Evaluation Metrics.** As the proposed SHaRe framework focuses on recommending Top-N items, we use two relevancy-based metrics: Recall@10 and Precision@10, and one ranking-based metric: Normalized Discounted Cumulative Gain (NDCG@10). In the training process, we randomly pick one item that a user never interacted with as the negative sample. In the evaluation processes, we rank items among all the candidates rather than sampling items. For fair comparison and unbiased validation, we repeat all the experiments 5 times and report average results.

**4.1.4 Parameter Settings.** In our experiments, we tune hyperparameters of baselines to the best by grid search for fair comparison. The learning rate of all methods is 0.001. For the general parameters, we empirically set the $L_2$ regularization coefficient to $1e-4$, the dimension of embeddings to 64, and the batch size to 2048. We optimize all the methods with Adam optimizer. In addition, we perform an early stopping strategy to stop training if all the metrics on the validation data do not improve after 50 epochs. The hyperparameter sensitivities of SHaRe are investigated in Sec. 4.6.

## 4.2 Overall Performance Comparison (RQ1)

To answer **RQ1**, we compare the vanilla version of social recommendation models with the version enhanced by SHaRe. The overall comparison result is shown in Table 2. The improvement marked by ↑ is computed by using the difference

of performances to divide the subtrahend. From the experimental result, we can draw the following conclusions:

In comparison to their vanilla versions, all social recommendation models enhanced by our SHaRe framework exhibit notable improvements across three datasets. Particularly, our SHaRe framework manifests a significant average improvement on the Yelp dataset, with each metric showing an improvement surpassing 4%. Among the three datasets, the LastFM dataset records the most modest improvement from SHaRe. When examining the vanilla models, SHaRe distinctly bolsters the performance of both DiffNet and LightGCN+social on the LastFM dataset. It also enhances LightGCN+social and MHCN on the Douban and Yelp datasets, with the enhancement of LightGCN+social reaching a remarkable 14.147%. The minimal enhancement attributed to the SHaRe framework is DiffNet on both Douban and Yelp datasets. We believe the possible reason is due to the simplistic structure of DiffNet and the neglect of item latent factors during the learning of user embeddings, whereas other models aggregate the item latent factors.

## 4.3 Influence of Homophily Ratios (RQ2)

To answer **RQ2**, we conduct the experiment to investigate the influence of different graph-wise homophily ratios on the enhanced SHaRe-DiffNet and SHaRe-MHCN under the same settings on synthetic sub-graphs. As shown in Fig. 5 and 6, the results indicate that: (1) SHaRe-DiffNet and SHaRe-MHCN demonstrate better trends than vanilla DiffNet and MHCN as the graph-wise homophily ratio increases, which means SHaRe can maintain comparable performance on different graph-wise homophily ratios. (2) Meanwhile, the performances of SHaRe-DiffNet and SHaRe-MHCN outperform vanilla DiffNet and MHCN across various graph-wise homophily ratios.

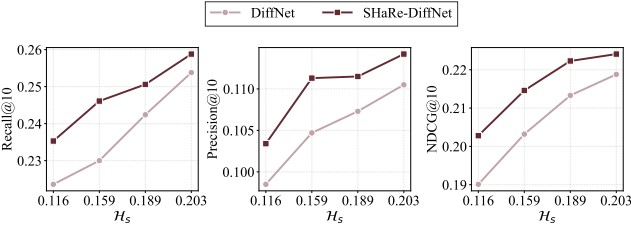

**Figure 5.** SHaRe-DiffNet results under different graph-wise homophily ratios.

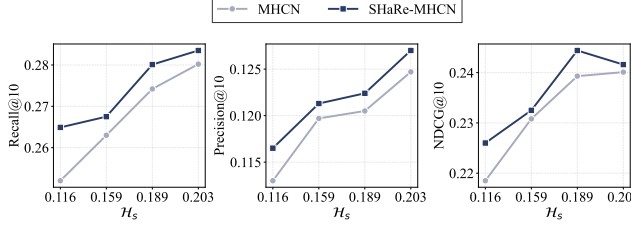

**Figure 6.** SHaRe-MHCN results under different graph-wise homophily ratios.

**Table 2.** Overall performance comparison of SHaRe on three datasets.

| Method | | LastFM | | | Douban | | | Yelp | | |
|---|---|---|---|---|---|---|---|---|---|---|
| | | R@10 | P@10 | NDCG@10 | R@10 | P@10 | NDCG@10 | R@10 | P@10 | NDCG@10 |
| DiffNet | vanilla | 0.1829 | 0.0826 | 0.1586 | 0.042 | 0.0728 | 0.0845 | 0.0462 | 0.0113 | 0.0296 |
| | **SHaRe** | **0.1904** | **0.0859** | **0.1646** | **0.0438** | **0.0742** | **0.0873** | **0.0472** | **0.0115** | **0.0303** |
| | | ↑4.101% | ↑3.995% | ↑3.783% | ↑4.286% | ↑1.923% | ↑3.314% | ↑2.165% | ↑1.770% | ↑2.365% |
| DiffNet++ | vanilla | 0.1864 | 0.0841 | 0.1584 | 0.0502 | 0.0832 | 0.0974 | 0.0490 | 0.0115 | 0.0308 |
| | **SHaRe** | **0.1906** | **0.0863** | **0.1613** | **0.0543** | **0.0852** | **0.0980** | **0.0501** | **0.0117** | **0.0319** |
| | | ↑2.253% | ↑2.616% | ↑1.831% | ↑8.167% | ↑2.404% | ↑0.616% | ↑2.245% | ↑1.739% | ↑3.571% |
| LightGCN+social | vanilla | 0.2019 | 0.0918 | 0.1756 | 0.0516 | 0.0877 | 0.1017 | 0.0582 | 0.0139 | 0.0375 |
| | **SHaRe** | **0.2102** | **0.0948** | **0.1816** | **0.0589** | **0.0945** | **0.1107** | **0.0642** | **0.0151** | **0.0422** |
| | | ↑4.111% | ↑3.268% | ↑3.417% | ↑14.147% | ↑7.754% | ↑8.850% | ↑10.309% | ↑8.633% | ↑12.533% |
| MHCN | vanilla | 0.2154 | 0.0974 | 0.1855 | 0.0606 | 0.0904 | 0.1080 | 0.0620 | 0.0146 | 0.0394 |
| | **SHaRe** | **0.2199** | **0.0987** | **0.1894** | **0.0637** | **0.0933** | **0.1110** | **0.0646** | **0.0153** | **0.0410** |
| | | ↑2.089% | ↑1.335% | ↑2.102% | ↑5.116% | ↑3.208% | ↑2.778% | ↑4.194% | ↑4.795% | ↑4.061% |
| Average Improvement | | ↑**3.138%** | ↑**2.804%** | ↑**2.783%** | ↑**7.929%** | ↑**3.822%** | ↑**3.890%** | ↑**4.728%** | ↑**4.234%** | ↑**5.632%** |

## 4.4 Analysis of Social Graph Rewiring Strategies (RQ3)

To answer **RQ3**, we analyze the benefits of three different strategies of SGR: **multi-SGR**, **w/o Warm-up** and **SHaRe-MHCN**. Note that **multi-SGR** rewires the social graph in all training iterations of each epoch; **w/o Warm-up** starts to rewire the graph at the first epoch; **SHaRe-MHCN** starts to rewire the graph from the 10th epoch, and only rewires the social graph once at the start of each epoch. The analysis results are shown in Table 3, which are collected on an AMD Ryzen 3990X 64-Core CPU and an NVIDIA GeForce RTX 4090 GPU.

In Table 3, we show an inference of time analysis (seconds per epoch) and the performance of different SGR strategies. The results indicate that **SHaRe-MHCN** outperforms both **w/o Warm-up** and **multi-SGR** with faster running time. Since **multi-SGR** executes SGR in the training of each iteration, the computational overhead will increase. The time per epoch in the LastFM dataset is about 2 times higher than that of SHaRe-MHCN, while it is more than five times in the Yelp dataset. In addition, we investigate how the number of rewired edges changes with epochs under different strategies. In Fig.7, we find that **w/o Warm-up** causes a large number of edges to be added and cut in the social graph, which indirectly leads to affecting the performance of SHaRe.

## 4.5 Ablation Study (RQ4)

To answer **RQ4**, we conduct the ablation study in two parts of experiments to investigate the effects of different components in SHaRe and to investigate different operations of Graph Rewiring.

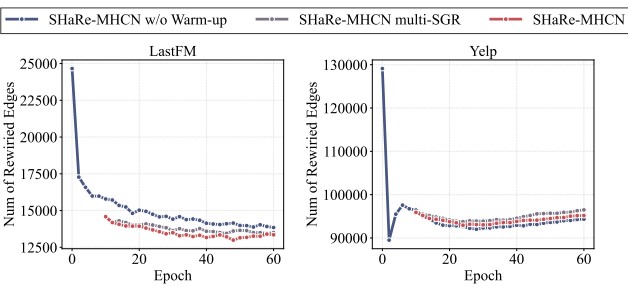

**Figure 7.** The number of rewired edges of different Social Graph Rewiring strategies.

### 4.5.1 Investigation of the Effect of SHaRe Components.
In this part, we investigate the effect of the component in SHaRe by removing it and retaining others to observe the differences in performance, we show the result in Fig. 8. Note that **w/o SGR** removes the Social Graph Rewiring component; **w/o HRA** removes the Homophilic Relation Augmentation; **w/o SW** replaces all the social weights $\widehat{S}_{(i,j)}$ in the rewired social graph with 1. From Fig. 8, we find that all these components are important for contributing to the effect of SHaRe. In the LastFM dataset, HRA is the most significant component for the improvement of SHaRe, social weight is the most significant component in the Douban dataset, and SGR is the most significant component in the Yelp dataset.

### 4.5.2 Investigation of the Effect of Social Graph Rewiring.
To investigate the effect of SGR, we analyze the impacts of edge adding and cutting operations. **Cut-only** variant only executes the cutting operation when performing SGR;

**Table 3.** Performances of SHaRe-MHCN with different rewiring strategies.

| Method | LastFM | | | | Yelp | | | |
|---|---|---|---|---|---|---|---|---|
| | R@10 | P@10 | NDCG@10 | Time | R@10 | P@10 | NDCG@10 | Time |
| multi-SGR | 0.2193 | 0.0991 | 0.1892 | 4.67s | 0.0635 | 0.0149 | 0.0398 | 63.10s |
| w/o Warm-up | 0.2176 | 0.0984 | 0.1882 | 2.22s | 0.0631 | 0.0147 | 0.0397 | 11.79s |
| **SHaRe-MHCN** | 0.2199 | 0.0987 | 0.1894 | 2.20s | 0.0646 | 0.0153 | 0.0410 | 11.69s |

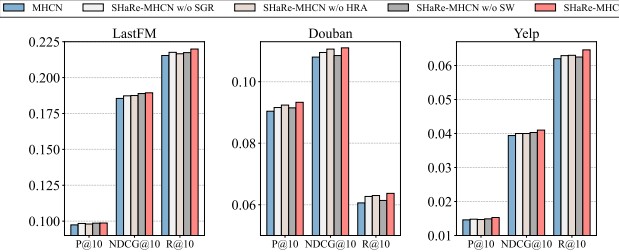

**Figure 8.** Investigation of the components in SHaRe.

**Add-only** variant only executes the adding operation when performing SGR. As shown in Fig. 9, both edge adding and cutting operations significantly contribute to the effectiveness of SHaRe. The contribution from edge cutting is greater than that of edge adding in the LastFM dataset, while in the Douban and Yelp datasets, edge cutting plays a more crucial role.

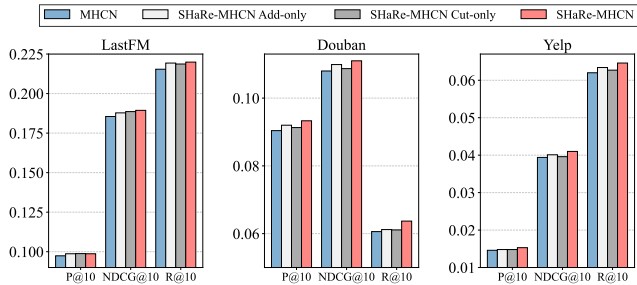

**Figure 9.** Comparison on different Social Graph Rewiring operations.

### 4.6 Parameter Sensitivity Analysis (RQ5)

To answer **RQ5**, we conduct the experiments to investigate the influences of two parameters in SHaRe: $\zeta$ for controlling the selection of positive samples in Homophily Relation Augmentation; $\lambda$ is for controlling the magnitude of contrastive learning loss. The results of sensitivity analysis are shown in Fig. 10 and 11. The range of $\zeta$ is $[0, 1]$, we conduct the sensitivity analysis of this parameter and set the values starting from 0.2 to 0.8, with a step size of 0.1. For $\lambda$, we report four representative values $\{1, 0.5, 0.1, 0.01, 1e-3, 1e-4\}$.

In Fig. 10, we observe that the performance of SHaRe is the best when $\zeta = 0.5$, $\zeta = 0.7$ and $\zeta = 0.3$ in LastFM, Douban

and Yelp, respectively. For $\lambda$, we find that SHaRe reaches the best performances when $\lambda = 0.01$, $\lambda = 0.5$ and $\lambda = 1e-3$ in LastFM, Douban and Yelp, respectively.

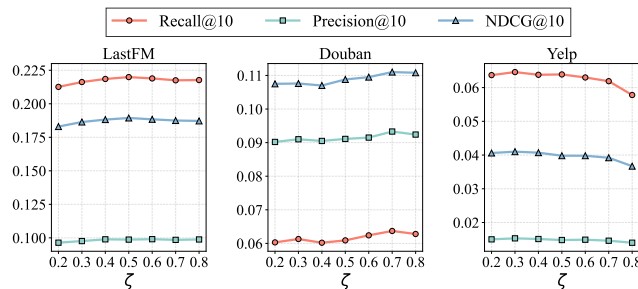

**Figure 10.** Influence of the sampling threshold $\zeta$.

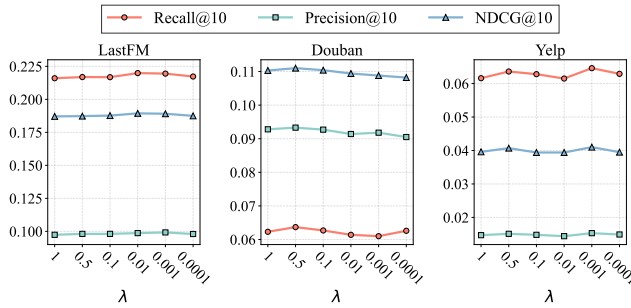

**Figure 11.** Influence of the contrastive learning coefficient $\lambda$.

## 5 CONCLUSION

In this paper, we explore the problem of low homophily in social recommendation and propose a general framework SHaRe that can be applied to any graph-based social recommendation backbone. The key enhancements to the backbone are the newly introduced Social Graph Rewiring and Homophilic Relation Augmentation. Social Graph Rewiring retains critical social relations while adding potential social relations which are beneficial for recommendations. Meanwhile, Homophilic Relation Augmentation refines homophilic social relations, enhancing the results of Social Graph Rewiring. We conduct extensive experiments on three real-world datasets, demonstrating the superiority of SHaRe.

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

## A RELATED WORK

### A.1 Contrastive Learning in Recommendation

Contrastive Learning (CL) has been prominently employed as an auxiliary task to address the data sparsity challenge inherent to recommender systems [27–29, 35, 39]. Wu *et al.* [29] apply CL to recommendation approaches by augmenting user-item bipartite graphs through the strategic removal of edges and nodes at a designated ratio. The subsequent objective is to maximize the consistency of representations learned from distinct views. Wei *et al.* [27] revisit the representation learning for cold-start items from an information theoretic perspective, aiming to maximize the interdependence between item content and collaborative signals, thereby mitigating the effects of data sparsity. Zhou *et al.* [39] adopt a methodology where attributes and items are randomly masked, facilitating sequence augmentation for sequence models pre-trained to maximize mutual information. Furthermore, based on the assumption of homophily in social graphs, Yu *et al.* [33] and Wu *et al.* [28] adopt different strategies, leveraging user social relations for data augmentation to capture homophily for CL.

### A.2 Graph Rewiring

Recent studies use Graph Structure Learning (GSL) methods such as Graph Rewiring (GR) to reduce bottlenecks in graph representation tasks, aiming to learn graph structures from original graphs or noisy data points that reflect data relationships [7, 15, 23, 37]. GSL methods aim to jointly learn an optimized graph structure and its corresponding node representations. Methods that change the structure of graphs to enhance performance for downstream tasks are often generically referred to as graph rewiring [2, 4, 8, 24]. For instance, in the extensive applications of GR, Bi *et al.* and Li *et al.* [3, 13] adopt GR methods to approach the low homophily problems in the classification of heterophily graph, Guo *et al.* design a GR method to handle low homophily problem of heterogeneous graphs [9].