# OpenReview forum: "Challenging Low Homophily in Social Recommendation"
_ACM.org/TheWebConf/2024/Conference — TheWebConf24_

### Official Review · Reviewer_NLK1 · 2023-11-14

**Novelty:** 6
**Technical Quality:** 4

**Review:**

The goal of this paper is to challenge the role of social homophily for the task of item recommendations. In particular, the authors propose that not all social connections are indications for homophily, and thus, when designing a recommendation system via propagating preferences over a social network, care should be taken to account for non-homophilous edges. Their assumption and empirical observation is shown to hold true on three popular datasets. To account for the non-homophilous edges and still be able to leverage social connections for item recommendations, the authors propose to rewire the social graph, so that low-homophily edges are discarded and (presumably) high-homophily edges are added, and then can apply any black-box social recommendation method. The rewiring task is accomplished via propagating item embeddings to users, and connecting the users based on those embeddings, and via contrastive learning to ensure that highly homophilous pairs of users are embedded to near-by data points. The proposed method is compared against black-box social recommendation baselines and it is shown to provide a small (but consistent) improvement.

Personally, I really enjoyed reading the paper. I found very interesting and convincing the main idea of the paper that not all social connections indicate homophily, and I considered sound the approach of the authors to improve social recommendations by rewiring the social graph.

At the same point, the paper has some limitations. In particular:

1. Beyond the main idea of the paper, the proposed method is technically not particularly deep and there is lack of underlying theory of the properties of the proposed method.

2. How is the proposed method different than standard (not graph based) collaborative filtering approaches, where user similarity is computed based on item similarity, and then recommendations are created by considering similar users? In such a case, social network information can also be incorporated to address the cold start problem. Should the authors had empirically compared their method against such approaches?

3. In the experiments, the improvements are relatively small. How significant are the results obtained by the new method?

**Questions:**

I would appreciate if you can provide answers to points 1, 2, and 3, above.

**Ethics Review Description:**

No ethics issues.

**Reviewer Confidence:**

2: The reviewer is willing to defend the evaluation, but it is likely that the reviewer did not understand parts of the paper

**Scope:**

4: The work is relevant to the Web and to the track, and is of broad interest to the community

---

### Official Review · Reviewer_BSS5 · 2023-11-20

**Novelty:** 6
**Technical Quality:** 7

**Review:**

This paper introduces a data-centric framework, namely SHaRe, designed to indirectly enhance social recommendation performance by improving the preference homophily of the social graph. SHaRe initiates graph rewiring by eliminating edges with low homophily scores and introducing potential edges with high scores. These scores are computed based on user embedding similarity. Leveraging the rewired graph, SHaRe employs contrastive learning to enhance correlations among user embeddings for users with high homophilic preferences. Notably, SHaRe is plug-and-play, allowing seamless integration with various social recommendation models. Extensive experiments conducted on three datasets with four baselines demonstrate SHaRe's effectiveness in improving the performance of these baselines.

In general, the paper is well-written and easily comprehensible. The technical details are clearly elucidated and logically sound. The experimental results are compelling. The strengths and weaknesses of this paper are outlined below.

Strengths:

S1. The novel and promising research direction of enhancing social recommendation by focusing on a "data-centric" method, rather than designing intricate neural blocks, is commendable. The intuition that cleaner and more reliable data can lead to a more effective model is well-founded.
S2. The proposed technique, SHaRe, is straightforward and technically sound. The presentation of technical details, including graph rewiring and graph contrastive learning, is clear. The inclusion of Algorithm 1 and the workflow figure enhances the understanding of the proposed method.
S3. The experiments are thorough, and the results are convincing. The ablation study analysis effectively demonstrates the impact and effectiveness of each proposed component.

Weaknesses:

W1. The rationale behind executing HRA on the original graph instead of the rewired graph is unclear. It would be beneficial to clarify whether this choice aims to defend against potential noise in the rewiring graph.

**Questions:**

Q1. Refer to W1.
Q2. What is the potential limitation of SHaRe? The limitations and potential future research directions are encouraged to be discussed.

**Reviewer Confidence:**

4: The reviewer is certain that the evaluation is correct and very familiar with the relevant literature

**Scope:**

4: The work is relevant to the Web and to the track, and is of broad interest to the community

---

### Official Review · Reviewer_SnFS · 2023-11-20

**Novelty:** 4
**Technical Quality:** 5

**Review:**

### Paper Summary
This paper investigated the low homophily ratio of user-user social connections in the user-item interaction graph. The paper proposed to use graph re-wiring approach to relearn useful social connections for recommendation.

**Pros**
1. The idea is interesting and the problem is important as there are noises from user-user social information for recommendation as there can be inconsistency in between.
2. The idea comes from the graph rewiring approach and enforces to align well with the graph recommendation task.
3. The paper story is straightforward with data analysis as support for motivation.

**Cons**
1. There are several missing literatures in this area, including [1, 2, 3, 4, 5, 6]. [5, 6] serve as strong baselines for fusing social information for recommendation.
2. The absolution improvements need to be further validated by statistical significance or standard deviation.
3. The method seems too similar with existing graph rewiring approaches.

[1]. UGSL: A Unified Framework for Benchmarking Graph Structure Learning

[2]. NodeFormer: A Graph Transformer for Node-Level Prediction

[3]. DIFFormer: Diffusion-based (Graph) Transformers

[4]. Graph Collaborative Signals Denoising and Augmentation for Recommendation

[5]. ConsisRec: Enhancing GNN for Social Recommendation via Consistent Neighbor Aggregation

[6]. Socially-Aware Self-Supervised Tri-Training for Recommendation

**Questions:**

* Can authors clarify what is the novel part other than the adoption of existing graph rewiring approaches?
* Can authors provide statistical significance or standard deviation as the improvements seem little?
* Can authors include more comprehensive discussion of existing graph structure learning methods?

**Reviewer Confidence:**

4: The reviewer is certain that the evaluation is correct and very familiar with the relevant literature

**Scope:**

4: The work is relevant to the Web and to the track, and is of broad interest to the community

---

### Official Review · Reviewer_RuaP · 2023-11-27

**Novelty:** 6
**Technical Quality:** 6

**Review:**

# Summary

This work focuses on the low homophily in social recommendation, particularly focusing on connections in the social graph. The authors argue that current social recommendation datasets exhibit low homophily, impacting the performance of existing methods like DiffNet and MHCN. The proposed method explicitly represents user preference similarity, enhancing homophily connections and demonstrating improved social recommendation performance in comprehensive experimental results.

# Pros:

P1. The studied problem is cutting-edge and practically significant. Homophily is a fundamental assumption in graph theory and motivates the proposed method well.

P2. The problem formulation is clear, and the presentation of proposed technical components is well laid out.

P3. The authors put forward a data-centric solution that can benefit various related methods.

P4. The evaluation part of the paper is sufficient and comprehensive.

# Cons:
C1.  The authors could additionally discuss how related social recommendation methods contribute to improving recommendation results and why the proposed method further enhances them.

C2. How do you calculate H_s using Eq. (1) when the soft edges are added by your method?

C3. Some references are missing, such as the information propagation layer and NDCG. The related work on homophily in graph representation learning and recommendations should be discussed with this work.

C4. A more detailed analysis of experimental results is needed. For example, in Table 3 and Figure 7, "w/o Warm-up" shows lower performance and a larger number of rewired edges. What is the reason behind these results?

C5. There are some typos in the manuscript:
1.	“we conduct experiments using two state-of-the-art models (SOTA).”
2.	“we show an inference of time analysis”.
3.	Top-𝑀 should be introduced first before the explanation.

**Questions:**

Overall this paper presents a quality solution to an important problem. My concerns are mainly associated with the lack of detailed explanation of concepts (C2) and discussions on the experimental results (C4).

**Reviewer Confidence:**

4: The reviewer is certain that the evaluation is correct and very familiar with the relevant literature

**Scope:**

4: The work is relevant to the Web and to the track, and is of broad interest to the community

---

### Decision · Program_Chairs · 2024-01-22

**Decision:**

Accept

**Comment:**

To alleviate the effect of low preference-aware homophily on social recommendations, the authors proposed a graph rewiring based approach to add highly homophilic social relations, while removing low homophilic relations in the social graph, and integrated a contrastive learning method to train the model. Experiments demonstrate the improved performance in different conditions (with varying homophily ratios). Overall, the idea is interesting and the paper is well-written. Reviewers suggested adding further clarifications/details (e.g., regarding some equations, algorithmic process, results significance, limitation discussion, underlying theory) and missing references. The novelty of the proposed method was also challenged by some reviewers. The authors are encouraged to carefully address the reviewers' comments in the final version if the paper is accepted.